# COVID-19 Risk Management and Stakeholder Action Strategies: Conceptual Frameworks for Community Resilience in the Context of Indonesia

**DOI:** 10.3390/ijerph19158908

**Published:** 2022-07-22

**Authors:** Iskandar Zainuddin Rela, Zaimah Ramli, Muhammad Zamrun Firihu, Weka Widayati, Abd Hair Awang, Nasaruddin Nasaruddin

**Affiliations:** Department of Agricultural Extension, Halu Oleo University, Kampus Hijau Bumi Tridharma, Kendari 93132, Sulawesi Tenggara, Indonesia; zaimahr@ukm.edu.my (Z.R.); mzamrun@uho.ac.id (M.Z.F.); wekawidayati@uho.ac.id (W.W.); hair@ukm.edu.my (A.H.A.); nasaruddin@uho.ac.id (N.N.)

**Keywords:** community resilience, coronavirus disease (COVID-19), livelihood, public health

## Abstract

The coronavirus disease (COVID-19) pandemic has affected people’s lives globally. Indonesia has been significantly affected by this disease. COVID-19 has also affected certain social and economic aspects of Indonesia, including community resilience. Through a variety of contexts and geographic locales, we explore the previously mentioned concept of resilience. From existing literature reviews, we develop a holistic framework for community resilience during the COVID-19 pandemic. Then, we formulate crucial factors for community resilience during the COVID-19 pandemic: natural capital, social capital, human capital, stakeholder engagement, community participation, technology, and communication. Strategic stakeholder action in the community resilience domain has facilitated increases in economic as well financial capital for adapting to and surviving deficits in productivity in the face of the COVID-19 pandemic. This study is a reflection on and a comparative review of the existing literature from different countries.

## 1. Introduction

Debates about community resilience and policy-making have increased significantly in recent years [1,2]. The concept has previously been found in diverse disciplines, such as natural sciences, management, economics, and psychology, and has started to influence regional sciences, planning theory, and practice [1,3]. The novel coronavirus disease (COVID-19) became a major epidemic threat to communities worldwide in December 2019 [4]. Besides the COVID-19 pandemic, the world has suffered various global public health threats such as HIV, the Influenza H5N1 virus subtype, SARS-CoV1, MERS-CoV, and Ebola [5]. COVID-19 can be easily transmitted through the mouth or nose of a person who has been infected or by touching surfaces infected with the disease [6]. Furthermore, many countries were not well prepared for this outbreak.

In Indonesia, the COVID-19 pandemic began on 2 March 2020, when an Indonesian citizen made direct contact with Japanese citizens who had been infected [7]. The infection rate of COVID-19 has significantly increased; from 3 January 2020 to 27 September 2021, there have been 4.2 million confirmed cases, with 3.36% resulting in death, as reported by WHO (2021). The capital city of Indonesia, Jakarta, had the most cases (20.4% of all cases), followed by West Java (16.7%) and East Java (9.4%) [7]. Although Indonesia has at least 134,207,308 doses of COVID-19 vaccines, this quantity is only enough to vaccinate 24.8 percent of the Indonesian population [8]. The COVID-19 pandemic has exacerbated Indonesia’s economy, de-industrialization, urban–rural disparities, inter-regional digital divide, unemployment, underemployment, the reduction in human resource development, and low engagement in global supply chains [9]. To overcome these exacerbations, the government and all parties involved must collaborate together; if not treated seriously, these problems will worsen human quality of life and public livelihood. This study critically discusses how to build community resilience and generates holistic recommendations and actions for stakeholders.

Building community resilience is challenging; collaborations that contribute to solving these problems are needed among parties, individuals, NGOs, institutions, and countries [10]. Cuello-Garcia et al. (2020) believed that social media has helped in handling the COVID-19 pandemic [11]. Therefore, our study examines how the COVID-19 pandemic has affected individuals and communities in terms of social, economic, environmental, and health changes in the context of Indonesia. This study then assesses stakeholders’ actions in risk management for community resilience. Finally, based on existing literature reviews, we develop a holistic conceptual framework for community resilience to the COVID-19 pandemic.

## 2. Effects of the COVID-19 Pandemic on Communities

The COVID-19 pandemic has had a serious impact on human health [12,13]. COVID-19 risk centration by Indonesian districts is concentrated primarily in high-population-density and urbanized areas [14,15]. Across Indonesia, the majority of COVID-19 cases are concentrated in Jakarta’s central, eastern, and western districts [7]. Furthermore, many positive cases tend to be located in suburban areas close to the city; in areas with high road and transport density; near trade, financial, and business centers; and near entertainment and food outlets [16]. Coelho et al. (2020) empirically demonstrated that areas with global connections, primarily the global air transportation network, is associated with the spread of COVID-19 [17]. However, tropical areas, which are located closer to the equator, can expect fewer new cases of COVID-19 infections [18].

The COVID-19 pandemic has altered many aspects of human life in Indonesia, including socioeconomic aspects. Fernandes (2020) showed that, as a result of the COVID-19 pandemic, 12 countries have experienced economic shock [19]. Hanoatubun [20] reported that the effect of the COVID-19 pandemic on the Indonesian economy is currently strongly felt within the community. For example, limited employment has hampered people’s means to satisfy their daily needs. This limitation is due to social restriction policies in areas where a high number of COVID-19 infections have been reported, which require people to stay at home and to not travel unless absolutely necessary. In 2020, the *New York Times* [21] reported that another impact of the COVID-19 pandemic is the opportunity for conflict at the community level, such as street demonstrations with an agenda of demands or insistence on the government to meet their basic needs. These demonstrations have the potential for social unrest, including looting, destruction, and violence, among fellow citizens. These kinds of events take place in a relatively short period. However, their impact can be just as destructive and traumatic as prolonged social conflict. The COVID-19 pandemic can also increase unemployment [22] and crime rates [23]. Furthermore, this pandemic has significantly caused changes in learning systems and patterns for students [24]. 

A direct impact of the COVID-19 pandemic is a decline in inbound tourists to Indonesia, which is, of course, not the only impact [25]. The current pandemic has affected the decisions of many investors in making influential investment actions [26]. In the agricultural sector, Aprilianti and Amanta (2020) reported that the COVID-19 pandemic is disrupting Indonesia’s food supply chain system. Agricultural employment is expected to drop by 4.87 percent, while domestic agricultural supply is bound to decline by 6.2 percent. Imports have fallen by 17.1 percent, while import prices have risen by 1.2 percent in the short term and will increase by 2.4 percent in 2022 [27]. In addition to the COVID-19 pandemic impacting the economic sector, it also triggers psychological disorders in adults and children, according to Cao et al. [28] and the United Nations Development Programme [29], and threatens the lives of people in many countries.

Furthermore, the COVID-19 pandemic has caused the amount of production to decline because many workers have been exposed to the virus. Additionally, logistics and transportation have become challenges because of the lockdowns in several countries. Accordingly, many countries must set strategies to ensure food supply throughout the pandemic and to control price inflation due to increased food purchases by the people [30]. Urban and rural economic activities and employment have also been affected. Therefore, with relatively limited socioeconomic capacity and capability, communities at socioeconomic disadvantages are highly vulnerable. In fact, Rogozhina [31] has provided proof of a rise in the unemployment rate as well as of a widening in socioeconomic inequality. In conclusion, the COVID-19 pandemic has significantly impacted various aspects of public life; health, economic, and social aspects, which have been affected by policies limiting the distribution system; the fields of education, tourism, exports, imports, and transportation systems; and the integration of all of these aspects. If issues in these aspects persist throughout this pandemic, then the sustainability of human life and the country’s resilience will come under threat. 

## 3. Conceptual and Analytical Frameworks

Holling [32] suggested the importance of understanding the ability of a system to manage or cope with change. Walker et al. [33] defined resilience as “the capacity of a system to absorb disturbance and reorganize while changing, to retain the same function, structure, identity, and feedbacks.” Berkes and Folke [34] stated that, in socioecological systems (SESs), humans must be considered part of nature; that is, the interdependence and coevolution of humans and nature exist at the individual and global scales Walker et al. [33] and Folke et al. [35] also established three different aspects of resilience: the capacity to sustain systemic shocks while preserving existing functions and structures; the capacity to face challenges, such as uncertainty and shock through renewal, reorganization, and learning in the current regime (adaptability); and the capacity to create a whole new trajectory rooted in radical changes in the nature of the system (transformability). These three aspects of resilience illuminate the need for diversity in actions at the individual, community, and institutional levels for the system to remain “dynamically stable”. Other studies, such as Aldrich and Kyota [36] showed that community resilience can increase social capital connections among individuals, thus allowing them to work collectively, to share norms, and to exchange information easily.

The concept of resilience is closely related to these SESs, which is defined as the “adaptive relationships and learning in social-ecological systems across nested levels, with attention to feedback, nonlinearity, unpredictability, scale, renewal cycles, drivers, system memory, disturbance events, and windows of opportunity” [37]. Thus, the authors proposed a socioecological and psychological developmental approach and a mental health approach to community resilience. These approaches emphasize community strength and build resilience through agency and self-organization, paying attention to people and their connections, values and beliefs, knowledge and learning, social networking, collaborative government, economic diversification, infrastructure, leadership, and views [37]. Community resilience is a concern for the government during disasters or major hazards, such as disease outbreaks. Furthermore, building a new capacity and spirit for affected communities is the most important action for the sustainability of future life. Wilson (2012) pointed out that the social, economic, and environmental aspects are essential variables in ensuring community resilience during environmental changes [38,39]. Meanwhile, Berkes and Ross (2013) asserted that resilience should refer to the concept of an SES [37]. This concept uses a socioecological and psychological developmental approach and a mental health approach to community resilience. Chen et al. [40] concluded that community resilience consists of economic resilience, institutional resilience, infrastructure resilience, environmental resilience, and social resilience.

According to SESs, resilience is the adaptation to changes caused by disasters. In terms of stakeholder engagement, the COVID-19 pandemic has undeniably impacted the lives of individuals. Thus, the dissemination of information and the inspection of individuals who have been infected to provide various solutions are crucial. The roles of individuals, groups, government agencies, and private institutions are indispensable. These individuals include doctors, nurses, local governments, the army, the police, social groups, and non-government organizations (NGOs). All of these parties have been working and collaborating to help victims of the COVID-19 pandemic. Wells et al. [41] stated that stakeholders (individuals, families, and agencies) in disaster planning contribute to community resilience. Stakeholder engagement at the local level in communicative actions is a crucial element of building community resilience to disasters such as academic–scientific entities, local councils, municipal services, rescue and emergency services, public–private entities, private social solidarity institutions, NGOs, and schools. Group stakeholders interpret and explore surface problems to identify fundamental problems and potential solutions. These intermediaries can be individuals who work with both scientists and end-users, and such organizations act as a bridge [42]. Furthermore, societal engagement involves direct and indirect interactions between social organizations and stakeholders-at-large, which involve the government, various institutions, and business establishments [43].

According to Rela et al. [44], the dimensions of community resilience are community adaptation, community action, and collective efficacy. Community action includes planning, accessing and using information, leadership, and connecting with outside organizations to leverage influence and effectiveness [45]. Collective efficacy is also a dimension of community resilience because agentic beliefs about producing the desired effects through collective action are essential for community action [46,47]. Meanwhile, Eversole [48] expressed that community action is based on inclusion, equity, social justice, human rights, and self-determination. Community resilience is achieved when the community works together to promote information and to control the spread of COVID-19. Community development organizations can use community cooperation and strategies to test the efficacy of various interventions on improving community resilience [49]. Furthermore, adaptive capacity in communities plays a vital role in the changes in social, economic, and environmental capital due to catastrophic disease outbreaks [50,51]. Community resilience can be realized by supporting mitigation strategies involving collective community action [52]. Government policy, community leaders, and social service agencies can have an impact on community action. If strategies are executed seriously, then community resilience can be achieved, such as the gradual reduction in various government regulations related to the spread of COVID-19.

### Community Resilience Factors

Technology and communication are important aspects of community adaptation [53]. A certain amount of knowledge and skills gained from various sources of information through mass media and direct dissemination increases the capacity of individuals and communities to protect themselves [54]. Furthermore, community resilience can be influenced by resources that operate through access to information [55]. Likewise, as Houston et al. [56] stated, a resilient community can rise up and adapt after a bad event [56]. Community resilience is generally considered a process indicated by society’s adaptation to a disaster or crisis. They use media and communication (communication ecology, public relations, and strategic communication) to review the dimensions of community resilience and to propose community resilience models.

In the dimension of economic and financial capital, adaptive capacity depends on certain assets, such as financial and natural resources, skills and opportunities, and livelihoods and lifestyle. Smit and Wandel [53] reported that one of the determining factors of adaptive capacity and community resilience is financial assets. Collier and Skees (2012) stated that overseeing financial policy implementation contributes to community resilience during a disaster [57]. Another aspect to note is that these risk management improvements can result in better financial performance, expand the reach of banking services, lower interest rates, and reduce volatility in credit access. Clarke [58] insisted that resilience must be improved by further legitimizing financial education efforts. Financial factors are also important in building community resilience amid disasters, such as an economic crisis in which the government should take note of the political economy and international markets [59]. 

Natural capital is the world’s stock of natural resources, which includes geology, soil, air quality, water, and all living organisms [60,61]. Some natural capital assets provide people with free goods and services, which are often called ecosystem services. These assets underpin our economy and society, thus making human life possible. Belle et al. [62] stated that natural capital is the environmental/ecological assets that a community is endowed with by nature. Other forms of natural capital include ecosystems, such as forests, rangelands, and mangroves. Natural capital indicators include air, land, and water quality; natural resources; biodiversity; scenery; topography; and location (proximity). The contribution of natural capital to community resilience serves as a foundation for the generation of other capital. It sustains all forms of life, promotes livelihood, provides protection against hazards, regulates climate, and protects the environment.

Social capital is the connection among people and organizations or the social cohesion that makes things happen in the community. Social capital indicators include trust, reciprocity norms, network structure, group membership, cooperation, sympathy, attachment, common vision and goals, leadership, depersonalization of policies, acceptance of alternative views, and diverse representation [62]. Social capital contributes to resilience because it aids in coordination and cooperation, facilitates access to resources, cushions against shocks [62], and fosters resilient communities [63]. Bonding and bridging, social cohesion, civic participation, heterogeneous socioeconomic relationships, political efficacy, and trust are factors of community health important for meeting the needs of the community [64]. The effects of bonding, bridging, and linking help community residents cultivate social capital to improve community resilience [65]. In this context, bottom-up, proactive, and collective community participation in planning and implementing activities alongside multiple stakeholders is required to recover from shocks [10,52,65]. Similarly, multiple stakeholders from within and outside the community should engage in these collaborations with the community [66]. By working with governmental policies and designing mechanisms and coordinated techniques to deal with a community risk, various stakeholders and community could further work together to take action. The risks and hazardous effects on the economy and businesses, the environmental ecosystem, human health and psychology, housing and amenities, networks and social connectedness, political and human conflict, and institutional change are summarized in Appendix A. As a result of major and minor shocks, these risks and hazardous effects will have impacts on individuals, families, communities, and the nation as a whole. Based on the above discussion, we summarize the main dimensions of community resilience and its contributing factors in Figure 1.

## 4. Community Risk Management

Social capital is the connection between people and organizations, or the social cohesion that holds societies together. Stakeholders are defined as individuals or groups with which businesses interacts who have a “stake” or vested interest in the firm. This stake is also described as a claim, interest, or right [67]. Freeman [68] defined stakeholders as “any group or individual who can affect or be affected by the achievements of an organization’s purpose”. Post et al. [69] stated that stakeholders are individuals and companies that contribute, either voluntarily or involuntarily, to its wealth-creating capacity and activities and are the potential beneficiaries and risk bearers. The types of stakeholders who can contribute to the resilience of the public in the case of natural disasters include the government, private organizations, NGOs, academics, and the community [10,70]. Table 1 shows that, in the case of the COVID-19 pandemic, all individuals, groups, and institutions can contribute to building individual and community resilience through collaboration [71], and other community involvement or actions can be enacted: cross-disciplinary and coordinated efforts; incremental efforts taken by HR, such as providing group instructions supported by reliable resources; public authority being enforced when the government enforces a policy to halt the spread of infections by the coronavirus disease; the media educating the public by effectively using the stay home hashtag; non-government organizations providing protective gear to front-line workers; and different associations being empowered to organize gatherings, pledges, and opportunities to provide necessities to emergency clinics and to coordinate efforts with medical service providers [72]. Some stakeholder actions for community resilience are presented in Table 1.

## 5. Stakeholder Action Strategy

Based on Appendix A, we attempt to recommend several stakeholder actions as a response to the risk of COVID-19 infections, especially in Indonesia, based on the dimensions of the community resilience framework formed. Table 1 and Figure 2 show that each dimension is set for a target stakeholder, followed by a description of that stakeholder action. Social capital (SC) refers to stakeholder social sensitivity, such as collaborating, helping, establishing relationships and mutual trust, leading, and being sympathetic. SC is an important factor for community health in meeting the needs of community resilience [62,64]. SC provides several benefits during crisis scenarios, and communities with high social capital respond more effectively than those with low social capital; social capital can assist in the recovery of the COVID-19 pandemic [78]. Evidence is growing that outbreaks such as the COVID-19 pandemic are well handled in places where social capital is high and communities are able to respond better to outbreaks [79]. SC is an important factor for helping stakeholders build community resilience in the case of the COVID-19 pandemic. As illustrated in Table 2, stakeholders can take actions such as (1) creating environmental sensitivities in order to feel a sense of belonging to a community, (2) intensifying commitment individually and in groups for the welfare of society, (3) facilitating people’s optimism about the future, (4) enforcing fair treatment regardless of background, and (5) maintaining distance when communicating to comply with health protocols. Stakeholder engagement (ES) refers to the involvement of other parties either individually or in groups in a project or activity to achieve the goals. Burnside-lawry and Carvalho [10] argued that the stakeholder approach builds community resilience from awareness to implementation. The expected ES requires cooperation and collaboration to solve problems. In Indonesia, the COVID-19 pandemic is handled by the Indonesian, provincial, and district governments. Assistance involves handling basic foodstuffs provided by the government and the private sector and distributing aid to the local community. Furthermore, local governments, such as at the sub-district level and below (Dusun), RW, and RT (neighborhood unit), should also assist the government in distributing aid [80]. The government has also issued several regulations regarding technical policies for handling the spread of COVID-19 at the central and district government levels and the roles of stakeholders. Djalante et al. [5] reported that the government needs to encourage coordination among, collaboration among, and utilization of all polymerase chain reaction (PCR) laboratories (research institutes, universities, hospitals, clinics, and the local government) to support PCR-based diagnoses. More attention must also be paid to health infrastructure development, including health laboratory development, in all provinces. Furthermore, the central government should implement a social distancing policy strategy [5,81]. 

At the national level, the government established the Indonesian Task Force for COVID-19 Rapid Response in March 2020 [5]. In the face of disasters, committed political and local leaders, along with strong and quality local leaders, are critical for setting goals and priorities to minimize the impact of crises [39,82,83]. Community confidence and trust in leaders, as well as leaders’ competence in risk and disaster management can reduce the negative impact of disasters [84,85,86]. Anti-vaccination sentiments have fueled concerns about vaccine safety, ineffectiveness, and side effects and about noncompliance with Islamic law (Halal) or non-Halal compounds [87]. The role of federal, provincial, and district governments, as well as the media in convincing the public of the benefits and safety of vaccines, along with procuring and delivering them in an efficient and equitable way, is critical for the COVID-19 crisis. The government has engaged Muslim leaders at both the national and grassroots levels, and the Indonesian Ulama Council has issued Halal certifications for the Sinovac and AstraZeneca vaccines [87]. Collaborations between UNICEF and Nahdlatul Ulama (Indonesia’s largest Muslim organization) have facilitated the acceptance of COVID-19 vaccines and have contributed to increased vaccination recognition among several prominent religious scholars in East Java [88]. The current government has also allocated additional funds to the Prosperous Family Programme (Program Keluarga Harapan—PKH), a Social Safety Net (SSN) program. This program benefits over 10 million communities at socioeconomic disadvantages, including pregnant women, children in early childhood, and people with disabilities. Additionally, electronic food vouchers are being distributed to nearly 20 million households [89].

Community participation (CP) refers to the participation of community members in developmental programs. CP is an important factor in building community resilience. Community involvement and participation are indicators that foster community resilience [90,91,92]. Yuda et al. [89] discovered that community leaders decided and participated in allocating community budgets to those who are vulnerable, namely families with lower socioeconomic statuses and migrant workers who have lost their employment. In handling the COVID-19 pandemic, CP can be a solution to minimizing the aftershock of the pandemic. Individuals infected with COVID-19 are detected through extensive testing. Prevention can be ensured through mass awareness campaigns and by protecting older adults and people who are more vulnerable, by isolating patients infected by COVID-19, by diagnosing rapidly, and by tracing patient contacts for quarantine and follow-up. Community support and government participation play important roles in controlling the spread of COVID-19 [93,94]. Residents of a village in South Kalimantan Province, for example, offered quarantine homes for residents diagnosed with COVID-19 as well as a community ambulances [89]. The community also encouraged education programs to support students in raising awareness about COVID-19 through online learning. Additionally, family and community support contributed to community resilience [95].

Information and communication technology (ICT) refers to how stakeholders use media to guard against misinformation during the COVID-19 pandemic. The Indonesian Ministry of Health provides real-time data on COVID-19 on an established website, https://www.covid19.go.id/ accessed on 1 July 2021, which contains various coordinated information [5]. Stakeholders can use the media to campaign about the dangers of COVID-19 from an early stage and to convey various policies and technical procedures so that the public does not become exposed to infections. ITC is the best solution for the government and organizations as a COVID-19 management strategy [96,97]. An organization can better plan for handling COVID-19 infections quickly and in any situation; ICT helps keep the environment sustainable by incorporating efficient renewable energy [98]. ICT, such as telemedicine and virtual care, has greatly assisted with the remote care of patients during the COVID-19 pandemic [99]. Elson et al. [100] reported that virtual care services using ICT could provide various non-dispensing functions; doctors can provide quality medical care services during the COVID-19 pandemic. Based on these findings, ICT has an important role, and the ICT dimension can be a tool for stakeholders in overcoming the problem of handling the COVID-19 pandemic. The main action that stakeholders can take is building trust in the community such that the information conveyed provides the correct solution. The contents of the message and the use of the right media are important factors. Furthermore, the government and other parties can provide accurate, accurate, and useful information. The widespread and continuous provision of COVID-19 information in mainstream conventional media, primarily television channels, radio stations, and newspapers, as well as in cyber media has created awareness and has effectively educated communities about the pandemic [5]. However, myths, rumors, incorrect beliefs, and conspiracy theories concerning COVID-19 vaccination have also circulated on social media. Instead of critiquing and re-examining the governments’ handling of the COVID-19, many local media serve as government–public liaisons, promoting authorities’ voices, educating about COVID-19 trends, and implementing strategies to intensify community awareness and action [101].

Economic and financial capital (EFC) refers to financial and natural resources, skills and opportunities, and livelihoods and lifestyles. This dimension is important in community resilience [1,53,57,90,102,103,104,105]. In handling the COVID-19 pandemic, EFC has been the main factor in helping people affected by COVID-19 infections, and the COVID-19 pandemic has had a negative impact on the economy around the world. Economic availability can help prevent the risk of an economic crisis [106]. The halt in developmental progress because of the COVID-19 pandemic has become a crisis, particularly in Europe [107]. Similarly, in Indonesia, where the economy collapsed due to the COVID-19 pandemic [26], the EFC is an important factor in maintaining societal resilience and state recovery. Providing cash in exchange for work in local villages has been implemented as a recovery action for vulnerable and low-income Indonesian communities [5]. Furthermore, in order to survive, the small-business community has gradually migrated from conventional business operations towards the use of the Internet of Things (IoT) and smartphones [108]. Indonesia’s corporate social responsibility (CSR) actions are more focused on the following: Empowering vulnerable workers; micro, small, and medium enterprises (MSMEs); housewives; and youth;Mentoring about socialization, healthy lifestyles, community social–psychological health, and the mitigation and preservation of the environment; andProviding aid to students, parents, and teachers [109].

Natural capital (NC) refers to providing people with free goods and services, often called ecosystem services. NC includes geology, soil, air quality, water, and all living organisms [60,61]. These factors are essential in building community resilience [50,110,111,112,113,114]. NC has become important in resuming normal life and developmental processes following the COVID-19 pandemic [115]. NC can support and promote various resources to ultimately support this process during the COVID-19 pandemic. The availability of natural resources will help in the recovery of affected communities [116]. Addressing the challenges of COVID-19 can be linked to sustainable food production by investing in increasing natural capital to increase productivity and resilience [117]. However, long-term recovery will require large investments in physical, human, and natural capital. Natural and human capital support has greatly assisted in the recovery of communities following the COVID-19 pandemic [118]. Given the importance of NC, stakeholder actions should be taken to manage NC by monitoring the availability of natural resources and the logistical distribution of food availability in various areas prone to the spread of COVID-19 and by safeguarding reserves in mineral and energy resources.

## 6. Conclusions

This study analyzed the resilience of communities in the face of the COVID-19 pandemic. Strategies related to strengthening the community’s resilience are crucial to developing and developed countries in today’s world. Resilience is the ability of individuals and communities to deal with change using available resources to improve life. People can face changes amid disasters, such as the COVID-19 pandemic. The factors of community resilience, namely, natural capital, human capital, social capital, stakeholder engagement, community action, technology and communication, and economic and financial capital, foster resilience, the capacity to absorb disturbances, competence, adaptiveness, and effective collaboration at the recovery stage in the face of changes. Thus, this resilience has been set as a goal for stakeholders (governments, corporations, NGOs, and the community). They must help the entire nation build up community resilience. However, there are some limitations and threats to the community resilience strategic action. Indonesia is segmented into islands, and remote communities have their own beliefs about COVID-19. The spread of anti-vaccination sentiments must be monitored and controlled via social engagement, by providing accurate knowledge, and by swaying beliefs about vaccination. Site visits and face-to-face engagements are the most appropriate actions in certain remote areas with limited ICT infrastructure and health facilities. To expand and upgrade these basic ICT infrastructures, health facilities, and advanced medical equipment, progressive investments are required. Huge public investment and funding are thus required from multiple local and international stakeholders to support community recovery from the COVID-19 pandemic. Analyzing community resilience is vital because human actions are always dynamic, and disasters always have the potential to occur anytime and anywhere. Natural disasters, including earthquakes, tsunamis, and hurricanes, pose such risks to communities. Climate change has also caused increases in sea level, flash floods, and droughts, all of which can lead to societal crises. Thus, the conceptual framework discussed in this study can serve as a guide in dealing with these disturbances.

## Figures and Tables

**Figure 1 ijerph-19-08908-f001:**
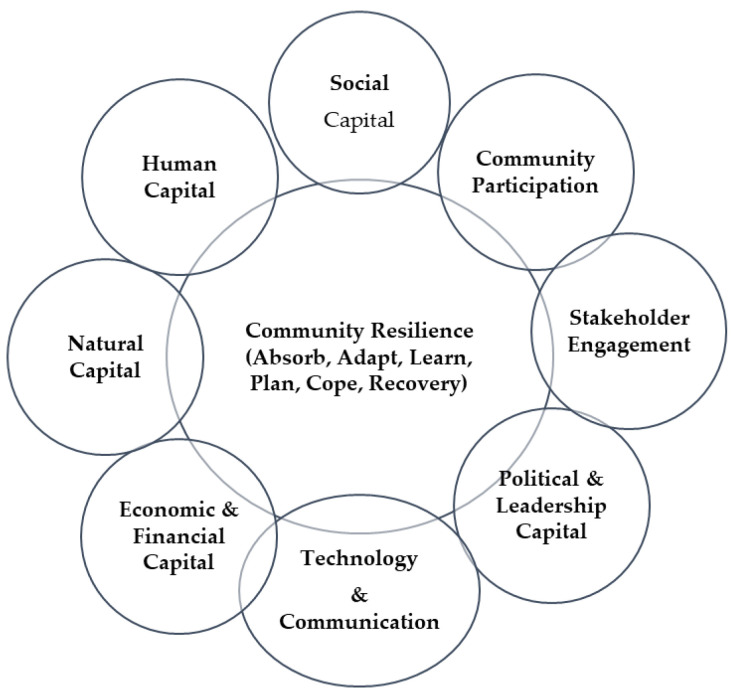
Factors of community resilience.

**Figure 2 ijerph-19-08908-f002:**
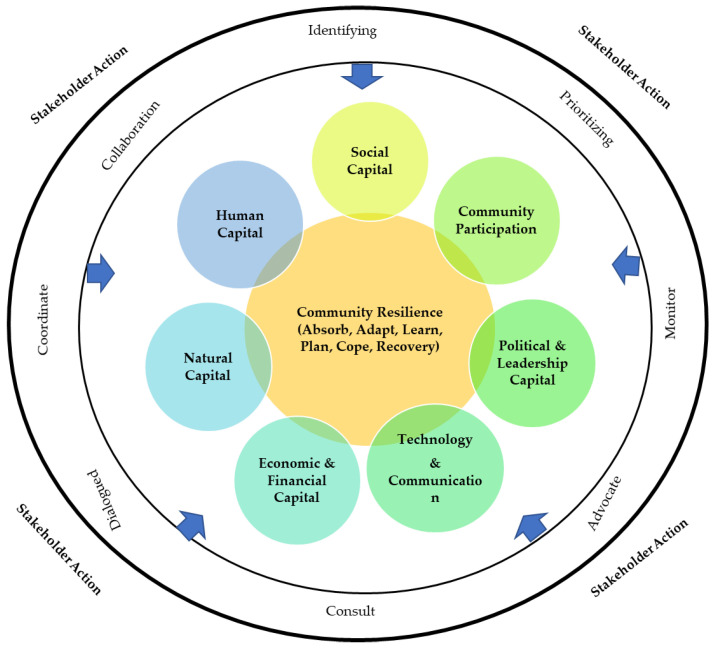
Stakeholder actions for COVID-19 risk management.

**Table 1 ijerph-19-08908-t001:** Stakeholder actions for community resilience.

Author	Issue	Type of Stakeholder	Stakeholder Action for Community Resilience
Larsen R.K. et al., 2011 [73]	Post-disaster recovery	Persons, groups, institutions, and the government	Mobilizing its agency by mobilizing social relations related to resource rights and access Supporting adaptive approaches that respond to the uncertainty of hazards and risks and to unstable conditions Enhancing the role of stakeholder institutions and processes through which a legitimate vision of resilience is generated
McKnight B. et al., 2016 [74]	Natural disasters	Non-profit companies	Companies strategizing around disaster response and presenting a coherent typology of non-profit corporate responses to natural disasters Developing assertive responses to natural disasters at the company (e.g., disaster recovery) and community levels
Burnside-Lawry J. et al., 2016 [10]	Disaster risk	Academic–scientific entities	Developing a set of studies about risk and vulnerability assessment
		Local councils	Promoting contact with local associations and citizen groups to increase their awareness
		Municipal services	Cooperating and collaborating with the campaign team
		Rescue and emergency	Providing data about disaster losses and support
		Public–private entities and private social solidarity institutions	Providing stakeholder training and awareness for first aid and drills
		NGOs	Supporting public awareness initiatives and providing free training to campaign stakeholders
		Schools	Organizing training and awareness activities about risk and disasters
Ashmawy I.K.I.M., 2021 [70]	Post-disaster	Governments, private sectors, and NGOs	Increasing the work of all parties, fostering relationships within the community, and improving their performance and work Increasing social and economic capital, infrastructure, and convenience, including schools
Cuello-Garcia C. et al., 2020 [11]	COVID-19	Health stakeholder	Being active and visible on social media Connecting with the social media channels of journals related to publications, reacting and becoming involved in relevant messages and discussions, and following relevant experts who are active on social media
Shah A.U.M. et al., 2020 [72]	COVID-19 outbreak	Government	The government imposing movement control orders to break the chain of COVID-19 infections and instructing the media to actively spread the stay home hashtag; non-governmental organizations, as well as convicts, producing personal protective equipment for front-line workers Encouraging cooperation with other organizations in organizing fundraising events to provide basic necessities, especially for hospitals Collaborating with health care providers
Fletcher F.E. et al., 2020 [71]	COVID-19	Community	Understanding the importance of cross-disciplinary expertise and collaboration Increasing human resources through community education and outreach by trusted sources
Zeneli A. et al., 2020 [75]	COVID-19	Government (nursing organizations)	Carrying out personnel distribution, reorganizing and maximizing nursing workflows, increasing new skills and knowledge, creating effective communication strategies, optimizing infection control policies, improving risk assessment and monitoring programs, and providing continuous personal protective equipment Strengthening the role of nurses as patient and caregiver educators, who are needed to promote infection-prevention behavior in the general population
Holtmann, G. et al., 2020 [76]	COVID-19	Stakeholder	Proactive planning with the involvement of relevant stakeholders in dealing with various scenarios, such as the economic and social impact of a pandemic
García-Sánchez L.M. et al., 2020 [77]	COVID-19	Private sector	Protecting the interests of shareholders and investors Supporting the welfare of the Spanish people in general and vulnerable groups in particular Combining previous altruistic actions with commercial interests

**Table 2 ijerph-19-08908-t002:** Recommended stakeholder action strategies for COVID-19 risk management.

Strategy	Stakeholder	Stakeholder Action
Social Capital Advancement	Community leader Local authorities Individual level	Enhance environmental sensitivity Increase commitment individually and in groups Facilitate the people around us Provide fair treatment regardless of background Maintain social distance when communicating Comply with health protocols
Stakeholder Engagements	Local authorities Government Non-government Community member Private industry	Coordinate the handling of the COVID-19 pandemic Collaborate in COVID-19 control tasks Promote awareness campaigns or advertisements about COVID-19
Community Participation	Community leader Community member	Support government programs Communicate with leaders for community safety Collaborative with social organizations Support local leaders Prioritize future community welfare Strengthen the skills and resources for effective plans
Inclusive Technology and Communication	Government Non-government Community Media industry	Organize discussions with other parties Establish a mechanism for accurate information dissemination Identify reliable individuals/organizations to provide information Ensure that local media provides accurate information Increase community trust in government officials
Economic and Financial Empowerment	Non-government organization Government Private industry	Prepare technical and non-technical services Guide the victim in solving the problem effectively Facilitate the community in dealing with risks Prepare resources for the community
Natural Capital Mitigation and Conservation	Government Community member Private industry	Ensure the availability of natural resources Ensure logistic availability for distribution Maintain the food supply, such as agriculture and fisheries Ensure the viability of mineral and energy resources

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
