# Peer review of "COVID-19 Risk Management and Stakeholder Action Strategies: Conceptual Frameworks for Community Resilience in the Context of Indonesia"

_ijerph, 2022, doi:10.3390/ijerph19158908_

Round 1
Reviewer 1 Report
Thank you for the possibility to review the paper. It is a very interesting and promising work presenting an overview of community resilience concepts with the COVID-19 in the background. I find this research as very useful and worth publishing, as I assume it can be broadly used to analyze different risks. Nevertheless, I suggest some minor corrections and additions, before it gets published. I am looking forward for a revised version of your paper.
Below you can find some my more or less general remarks.
Abstract, lines 17-18: there is something wrong or missing in this sentence (“Using this formula…”). Please correct it.
Line 32-33: can you give a reference for that? Even a press release would be fine here.
Lines 110-115: this idea of three aspects of resilience has been also operationalized and used in a research (doi.org/10.3390/su14042052) on local communities facing external shocks (natural disasters), and called three types of resilience (resistance, recovery and creativity) together forming community resilience arrangements. I think it is worth mentioning in the paper here or somewhere later.
Lines 170-171: Only in Indonesia this can influence community action? In my opinion you are free to tell that more broadly.
From line 174 you are describing the elements of community resilience. I suggest you to separate a subsection (e.g. 3.1. Community resilience factors). It improves the readability of this part of the text.
I am very curious where do you allocate institutional and organizational aspects that may influence community resilience. As I understood it is a part of social capital. If this is right, you should elaborate on this issue a little bit more, so for a reader it is clear, how institutions (formal and informal) and organizations affect resilience.
Figure 1: I have a suggestion to not to mention in the largest circle the words in brackets “Absorb, Adapt…”. Or instead add there as well “Learn”. But this remark is not crucial. It is just in case someone might feel that there is something missing.
Line 232: After “Social capital” there is missing something – maybe “which”?
Line 260-261: It is repetitive, you have already proven that social capital affects resilience.
Figure 2: For me it is not clear enough and hard to read – please put the stakeholder actions in circles (like the factors), and the arrows suggest that a particular action is connected with a particular factor of community resilience (e.g. Identifying and Social & Cultural Capital). If this was not your intention then please rethink this figure.
Table 2: It is very good. I would suggest to add references in the table and shorten the preceding paragraphs. It is not a crucial remark, just something to think of.
Line 386: Should not be it “Conclusions” instead of “Concluding”?
Line 387: This study does not discuss resilience of a community, it rather analyses community resilience in the face of COVID-19.
Line 399-400: Your study has a potential to be applicable in case of other risks, not only the COVID pandemic. You should elaborate on this a bit more, not just leaving one sentence at the end.
Author Response
We attached the revised file according to your comments.

Reviewer 2 Report
The article deals with a very interesting and topical issue. Although the authors rely on the method of analyzing literature (and not quantitative or qualitative research), a broad review of this literature, its ability to analyze it and formulate logical and rational conclusions make the article valuable. However, a few shortcomings need to be corrected:
1. You need to precisely indicate the purpose of the article, because at the moment it is unclear - I do not know if it applies only to Indonesia or the whole world.
2. The title also requires modification. The considerations in the article and examples of activities relate to Indonesia. It should be clear from the title.
3. I have the impression that references to literature are sometimes made incorrectly. I have marked a few such places in the text.
4. The work contains a lot about the opportunities and strengths of organizations and institutions operating in specific areas, but there is no (even brief) discussion of the limitations and threats of these entities.
Other issues requiring improvement / modification or clarification have been marked in the text as comments.

Author Response
We attached the revised file according to your comments
